# Pharmacoresistant Epilepsy in Childhood: Think of the Cerebral Folate Deficiency, a Treatable Disease

**DOI:** 10.3390/brainsci10110762

**Published:** 2020-10-22

**Authors:** Sarah Mafi, Cécile Laroche-Raynaud, Pauline Chazelas, Anne-Sophie Lia, Paco Derouault, Franck Sturtz, Yasser Baaj, Rachel Froget, Marlène Rio, Jean-François Benoist, François Poumeaud, Frédéric Favreau, Pierre-Antoine Faye

**Affiliations:** 1CHU de Limoges, Service de Biochimie et Génétique Moléculaire, F-87000 Limoges, France; sarah.mafi@chu-limoges.fr (S.M.); pauline.chazelas@chu-limoges.fr (P.C.); anne-sophie.lia@unilim.fr (A.-S.L.); franck.sturtz@unilim.fr (F.S.); yasser.baaj@chu-limoges.fr (Y.B.); frederic.favreau@unilim.fr (F.F.); 2CHU de Limoges, Service de Pédiatrie, F-87000 Limoges, France; cecile.laroche@chu-limoges.fr (C.L.-R.); rachel.froget@chu-limoges.fr (R.F.); 3CHU de Limoges, Centre de Compétence des Maladies Héréditaires du Métabolisme, F-87000 Limoges, France; 4Faculté de Médecine, EA6309 Maintenance Myélinique et Neuropathies Périphériques, Université de Limoges, F-87000 Limoges, France; francois.poumeaud@gmail.com; 5CHU Limoges, UF de Bioinformatique, F-87000 Limoges, France; paco.derouault@chu-limoges.fr; 6CHU Limoges, INSERM CIC 1435, F-87000 Limoges, France; 7CHU Necker, Enfants Malades, Paris, APHP, Service de Génétique, F-75743 Paris, France; marlene.rio@aphp.fr; 8CHU Necker, Enfants Malades, Paris, APHP, Service de Biochimie Métabolomique, F-75743 Paris, France; jean-francois.benoist@aphp.fr

**Keywords:** cerebral folate deficiency, *FOLR1* variant, neurodegenerative disorder, epilepsy, FRα protein crystallographic structure, pediatric

## Abstract

Cerebral folate deficiency (CFD) is a neurological disorder characterized by low levels of 5-methyltetrahydrofolate (5-MTHF) in the cerebrospinal fluid (CSF). The prevalence of this autosomal recessive disorder is estimated to be <1/1,000,000. Fifteen different pathogenic variants in the folate receptor 1 gene (*FOLR1*) encoding the receptor of folate α (FRα) have already been described. We present a new pathogenic variation in the *FOLR1* in a childhood-stage patient. We aim to establish the core structure of the FRα protein mandatory for its activity. A three-year-old child was admitted at hospital for a first febrile convulsions episode. Recurrent seizures without fever also occurred a few months later, associated with motor and cognitive impairment. Various antiepileptic drugs failed to control seizures. Magnetic resonance imaging (MRI) showed central hypomyelination and biological analysis revealed markedly low levels of 5-MTHF in CSF. Next generation sequencing (NGS) confirmed a CFD with a *FOLR1* homozygous variation (c.197 G > A, p.Cys66Tyr). This variation induces an altered folate receptor α protein and underlines the role of a disulfide bond: Cys66-Cys109, essential to transport 5-MTHF into the central nervous system. Fortunately, this severe form of CFD had remarkably responded to high doses of oral folinic acid combined with intravenous administrations.

## 1. Introduction

Folates or vitamin B9 (B9) are essential to many biological processes such as DNA synthesis and repair, regulation of gene expression, amino acids (AA) metabolism, myelin formation and neurotransmitters synthesis [1]. They are also well known to be involved in the neural tube formation during embryogenesis [2].

Although the fact that folates naturally exist in a variety of foods [3], deficiencies in humans can occur. They are absorbed in the intestine and then metabolized in the liver into 5-methyltetrahydrofolate (5-MTHF). Folate is a family of B vitamin found in human food as folinic acid (5-formyl-THF) converted into 5-MTHF or in synthetic folic acid supplementation. Folic acid is reduced to 7,8-dihydrofolate (DHF) and then into 5,6,7,8-tetrahydrofolate (THF) by dihydrofolate reductase (DHFR) [4]. A carbon unit is transferred from serine to THF by serine-hydroxy methyl transferase (SHMT). This reaction produces glycine and 5,10-methylene-tetrahydrofolate, which is involved in nucleic acid synthesis and repair. The latter product is then converted to 5-MTHF by methylenetetrahydrofolate reductase (MTHFR). Folinic acid can either be immediately available for metabolic processes or be converted into 5,10-methylene-THF consecutive activities of MTHF synthetase and MTHF-dehydrogenase [5]. During the conversion of 5-MTHF to THF, a methyl group is released. This group is used for the homocysteine remethylation leading to methionine which is the precursor of S-adenosyl-methionine (SAM), the universal one-carbon donor involved in the methylation of DNA, RNA, lipids, proteins and neurotransmitters [6] (Figure 1).

In the central nervous system, different folate transport systems are found, including the reduced folate carrier (RFC), the proton coupled folate transporter (PCFT) and the receptor of folate alpha (FRα). FRα represents the higher affinity folate transport system compared to RFC, and PCFT. 5-MTHF uptake into the brain is mainly mediated by FRα and PCFT in the basolateral membrane of choroid plexus epithelial cells and by FRα and RFC at the apical brush border [6]. The mechanism of these transporters involved in 5-MTHF flux across the blood-brain-barrier is not yet fully understood, but some mechanisms have been proposed for intracerebral transport. According to Grapp et al., and Zhao et al., 5-MTHF binds to FRα at the basolateral side, as it reaches the choroid plexus. The 5MTHF-FRα complexes are internalized through receptor-mediated endocytosis. After internalization, the resulting vesicles are translocated into GPI-anchored protein-enriched early endosomal compartments (GEECs). Then, GEECs are transferred to maturated endosomes that are multivesicular complexes (MVC). Intralumenal vesicles (ILV) containing FRα complexes accumulate within MVC and are released as exosomes at the apical side of the epithelium into the CSF, before being endocytosed again by ependyma cells [7,8]. According to Requena Jimenez et al., in this condition, 10-formyl-THF-dehydrogenase (FDH) is crucial, colocalizing with 5-MTHF and FRα within CSF endocytic vesicles. According to in-vitro studies, FDH binds with FRα and is involved in the folate transport regulation by controlling 5-MTHF fluctuations. FDH levels in CSF might also have a key role in the maturation of the leptomeninge arachnoid [9].

Cerebral folate deficiency (CFD) is a rare neurological syndrome (OMIM #613068) associated with low levels of 5-MTHF in the cerebrospinal fluid (CSF). Different causes altering the function of FRα have already been described such as mitochondrial disorders, *folate receptor 1* gene (*FOLR1*) mutations [10] and the potential existence of FRα auto-antibodies [11]. The *FOLR1* gene is located on chromosome 11 q13.4 and contains seven exons. At least fifteen pathogenic variants of this gene have been described, resulting in a defective protein [6].

## 2. Case Presentation

Ethics approval was obtained from the ethic committee of Limoges University Hospital: N 364-2020-20, as well as the consent of both parents to diffuse this data. This study was performed in accordance with the Declaration of Helsinki. A three-year-old girl was admitted to pediatric emergency room for a first febrile seizure episode. She is the only child of a non-consanguineous couple. She was eutrophic full-term born with a normal vitality index. Her psychomotor development was normal until the age of eighteen months, after which she exhibited ataxic gait, tremors, irritability, sleep disorders and speech difficulties. To treat this epileptic episode, a diazepam intrarectal injection was performed. Seizures were thought to be induced by a fever (38.5 °C).

A second episode of epileptic seizures occurred ten months later, followed by other convulsive episodes without fever in the next months. A brain MRI was then performed and revealed a severe hypomyelination, according to hypersignals in the subcortical white matter particularly in the cerebellar and subtentorial area predominantly posterior (Figure 2). Electroneuromyography did not show signs of peripheral nerve impairment. Chromosome analysis by karyotyping and array-comparative genomic hybridization did not detect any abnormalities. Hearing testing and ophthalmological examinations were also normal. Biological parameters were assessed in plasma (ammonia, lactate, pyruvate, creatinine phosphokinase, acylcarnitines profile, sialotransferrine profile, AA, phytanic acid, pristanic acid, very long chain fatty acids), in urine (AA, organic acids profile, sulfatides, mucopolysaccharides, leukocyte enzymes) and in CSF (AA). All previous parameters showed normal patterns.

At five years of age, an increase in seizure frequency was observed. Bilateral brain calcifications of globus pallidus and subcortical structures in parietal and occipital lobes were revealed by a cerebral CT-scan (Figure 3). Slow activity and a spike-and-wave pattern associated with an epileptiform activity, in particular in hypomyelination areas, were observed on an electroencephalogram. Different combinations of antiepileptic drugs were used in first as monotherapy and then as bitherapy including clobazam, sodium valproate, lamotrigine, levetiracetam, topiramate, piracetam and zonisamide. However, antiepileptic drugs were ineffective and side effects occurred, particularly with clobazam, topiramate and piracetam.

Dyskinesia, cerebellar atrophy and clinical manifestations of encephalopathy became progressively more severe. At about eleven years old, the patient’s speech abilities and motor function had gradually declined and her independent walking ability was lost. Myoclonic seizures were observed at least once a day and were associated with probable atonic seizures.

Extensive metabolic work-up was carried out. CFD was suspected after CSF analysis, which showed drastically low 5-MTHF levels (1 nmol/L; Ref. Values: >44 nmol/L) associated with an unusual decrease of folate levels in the plasma (<2 ng/mL; Ref. Values: 3.89–26.8 ng/mL) despite normal folate levels in red blood cells. A nutritional folate deficiency was firstly excluded because of normal plasma concentrations of homocysteine, vitamin B12 (B12), normal red blood count and normal folate levels in red blood cells supported by a balanced diet according to family information. Other metabolic investigations in CSF were not contributive (Table 1). A panel of genes involved in CFD by the NGS approach was analyzed (Appendix A). A library was obtained using the Sureselect strategy (Agilent) targeting exons and exon–intron boundaries (+/−50 bp). Sequencing was performed by a NextSeq500 (Illumina) sequencer. Coverage was 100% at a depth of 30× and the bioinformatic pipeline allowed SNV (single nucleotide variants) and CNV (copy number variation) detection. The diagnosis of CFD was confirmed by the detection of a new homozygous *FOLR1* substitution, c.197 G > A (p.Cys66Tyr) located in exon 4. No other pathogenic genetic variation was detected using an NGS-specific panel. As expected, both parents were heterozygous carriers of this variant.

The described variation in our case has never been reported in ClinVar and is found only once out of 251,476 alleles in the GnomAD database (version 2.1.1) [12]. Interestingly, the pathogenicity predictions assessed with common software gave a high pathogenic score for this variant (Table 2). Although this variant is not localized in the folate binding site sequence (Figure 4), different software predicted instability of the protein induced by the variant (Table 3). This variation results in a substitution of a cysteine to a tyrosine in the 66th position associated then with a disulfide-bond disruption between Cys66 and Cys109 probably leading to a modification of the FRα tridimensional conformation.

## 3. Discussion

We describe a new pathogenic variant (c.197 G > A, p.Cys66Tyr (NM_016724.2)) in *FOLR1*, associated with a severe form of CFD. As of today, reported variations in *FOLR1* have been splice-site, stop, missense mutations (most common) or duplications. These variations led to decreased protein expression and a loss of membrane localization of the FRα protein [24]. Grapp et al. [1], have suggested that missense mutations might lead to an instable conformation of FRα explained by the loss of a structurally important disulfide bond or disturbed post-translational processing. Splice and stop mutations might result, respectively, in a truncated FRα and in a premature termination codon and subsequent complete lack of FRα expression. Duplications are likely to cause major misfolding leading to premature degradation of FRα [25].

The human FRα crystallographic structure used in this study (PDB 4LRH) [26] includes 6 exons and 218 AA residues organized in 6 α-helices and 4 β-strands. It is unusually rich in disulfide-bonds with eight of them in this protein. Protein disulfide-bonds are formed by two oxidized cysteine residues which significantly stabilize the protein structure. Normal and stable FRα structure is essential for maintaining an effective binding site and allowing therefore a normal 5-MTHF transport capacity. In our case the mutation of Cys 66 to Tyr lead to the absence of disulfide bond which, in turn, is likely to significantly modify the conformation of FRα. This may lead to an increase of the two domains distance (Cys66 is located on the second β-strand (pos 63–67) and Cys109 is located on the fourth α-helix (pos 94–109) (Figure 4)). However, the impact to FRα function is not obvious since Cys66-Cys109 disulfide bond is located far away from the functional site. Thus, we hypothesize that the Cys66Tyr mutation mostly affects protein dynamics which in turn may modulate the function of FRα. Thus, this altered protein most likely leads to an interruption of folate transport into the brain and a depletion of 5-MTHF in the cerebral compartment [27,28].

According to Steinfeld R. et al. [25], a 5-MTHF deficiency in the brain leads to disorders of myelin synthesis, which could be explained by the brain specific biochemical pathway connecting folate to choline metabolism based on methyl group transfer. Indeed, after the release of its methyl group, SAM is converted to S-adenosyl homocysteine (SAH), which is a metabolic precursor of homocysteine. SAM regeneration is ensured by a methyl group transfer from methionine, previously formed by homocysteine methylation provided by the conversion of 5-MTHF to THF. Alternatively, the previous methylation can be obtained through the choline oxidation pathway by which a methyl group is released after converting betaine to dimethylglycine. This conversion is mediated by the betaine-homocysteine S-methyltransferase. This latter pathway is stimulated in CFD and particularly when 5-MTHF is decreased drastically in CSF, resulting in a secondary choline deficiency because of betaine formation (Figure 1). Since choline is required for the synthesis of phosphatidylcholine, phosphatidylinositol and sphingomyelin [25], a choline deficiency leads to brain white matter disruption and often to an inositol depletion. This stimulated process during 5-MTHF deficiency could explain neurological symptoms.

CFD typically occurs between 6 months and 4.5 years of age, with a median onset of 2 years [6]. In our case, the child’s neurological development was normal until the age of eighteen months. This symptom-free interval between birth and initial symptoms suggests the existence of an alternative transport pathway independent of FRα. A previously described hypothesis is related to the FRβ expression, which may compensate the loss of FRα function in the first years of life. Indeed, FRβ is highly expressed during the fetal and postnatal periods and it has an important homology with FRα (70% of AA identity) [25].

Supplementation with folinic acid has been reported to be efficient with a rapid reverse of neurological symptoms and an improvement of brain function. This molecule is known to pass across the non-specific pathways of PCFT and RFC directly into CSF where it is immediately available for folate metabolism [10,25,29,30]. It is established that the earlier treatment is administered the better the response is [6]. Moreover, it is important to underline that folic acid is not recommended in CFD treatment as folic acid strongly binds to FRα and inhibits its residual activity if any, limiting even more 5-MTHF transport to the brain [1]. In our case, an oral treatment with calcium folinate at 8.9 mg/kg dose was administered daily as soon as the diagnosis was established, based on the hypothesis that calcium folinate supplementation increases residual transport of 5-MTHF into CSF and CSF 5-MTHF levels. This treatment was combined with sodium valproate and zonisamide. Beneficial effects were observed during the second month of treatment with no side effects observed. According to the literature, these effects are due to a normalization of glial choline and inositol content [1]. In several studies [1,31,32], oral folinic acid combined to its intravenous administration has been shown to be more efficient than oral administration alone. Intravenous injections of calcium folinate were also administered to the patient once a week for three weeks then monthly. A significant increase of 5-MTHF blood levels, rising from 2 ng/mL up to 39.5 ng/mL, was noted within one month following treatment onset. After two months of treatment, significant improvements of social interactions and neurological symptoms were observed with a better motor coordination and myoclonic seizures frequency decrease without side effects observed. These results suggested that folinic acid is efficient with more than a 50% reduction in seizure rates [33]. Intravenous injections of calcium folinate were finally given at a 500 mg/week dose to optimize therapy. Cognitive functions, language skills and walking ability were improved underlining the role of an early diagnosis (Table 4).

## 4. Conclusions

This case highlights that CFD due to *FOLR1* gene variation c.197 G > A inducing p.Cys66Tyr substitution, is a rare but treatable neurodegenerative disorder of early childhood. CSF 5-MTHF levels should be assessed as soon as typical neurological symptoms occur during the first years of life to establish diagnosis and hence alleviate brain white matter defects. The study on the FRα conformation underlines the role of the cysteine residue (p.Cys66) on the protein tridimensional conformation in stabilizing this transporter and allowing efficient 5-MTHF transfer to the CSF.

## Figures and Tables

**Figure 1 brainsci-10-00762-f001:**
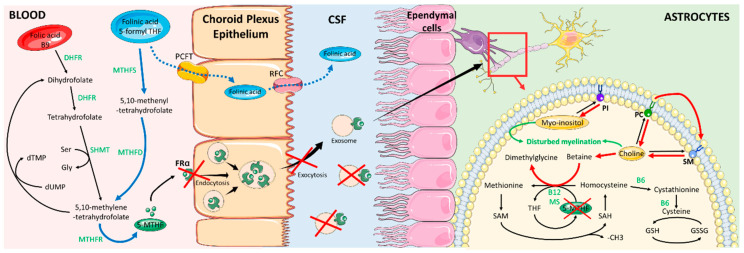
Folic acid metabolism and 5-MTHF transport across the choroid plexus epithelium in the brain. Red arrows and red crosses indicate the alternative pathway induced by FRα deficiency. Blue arrows indicate effects of folinic acid treatment. KEYS: 5-MTHF: 5-methylenetetrahydrofolate; B6: Vitamin B6; B12: Vitamin B12; CSF: cerebrospinal fluid; DHFR: dihydrofolate reductase; FRα: receptor of folate alpha; Gly: glycine; GSH: glutathione reduced states; GSSG: Glutathione oxidized states; MS: methionine synthase; MTHFD: methylenetetrahydrofolate dehydrogenase; MTHFR: methylenetetrahydrofolate reductase; MTHFS: methylenetetrahydrofolate synthetase; PC: phosphatidylcholine; PCFT: proton coupled folate transporter; PI: phosphatidylinositol; RFC: reduced folate carrier; SAH: S-adenosyl homocysteine; SAM: S-adenosyl-methionine; Ser: Serine; SHMT: serine-hydroxy methyl transferase; SM: sphingomyelin.

**Figure 2 brainsci-10-00762-f002:**
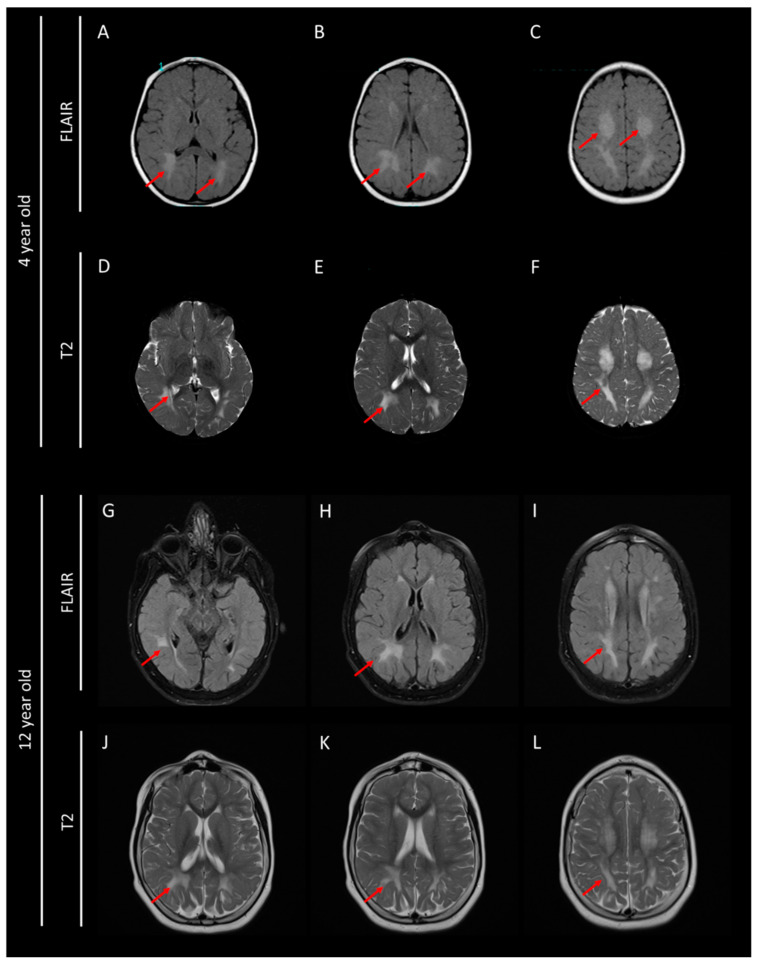
Brain MRI performed at 4 and 12 years old with FLAIR (**A**–**C**,**G**–**I**) and T2 (**D**–**F**,**J**–**L**) sequences. Red arrows show diffuse white matter abnormalities linked to an hypomyelination.

**Figure 3 brainsci-10-00762-f003:**
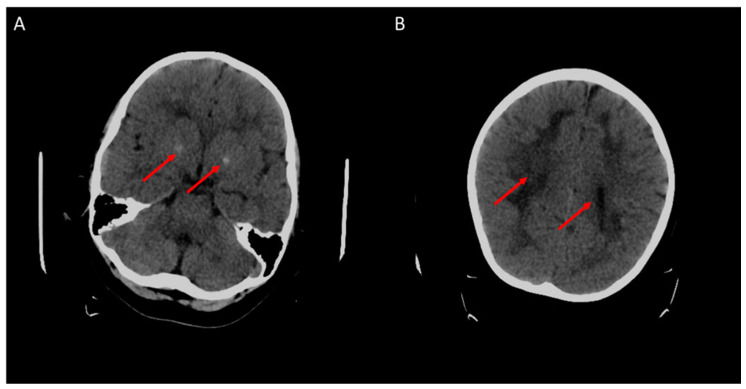
Cerebral CT-scan at 4 years old. Red arrows show brain calcifications (**A**) and diffuse white matter abnormalities (**B**).

**Figure 4 brainsci-10-00762-f004:**
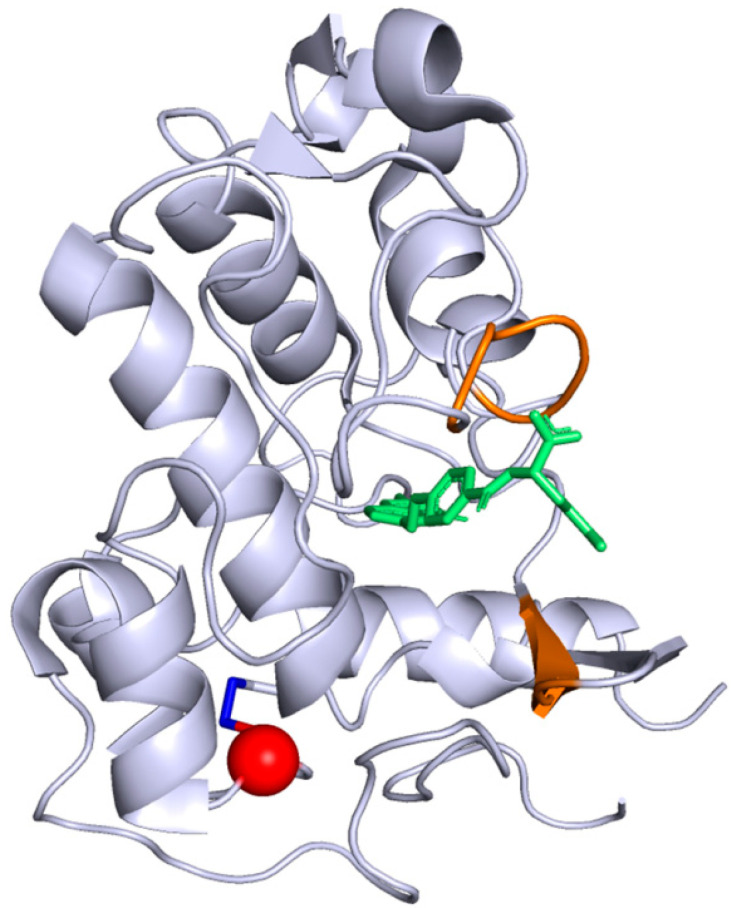
Crystallographic structure of FRα protein, from the Protein Data Band (4LRH). The folate is in green, the folate binding site is colored in orange. The Cys66Tyr substitution position induced by the pathogenic variant described in our case report is represented in red while the disulfide bond between Cys66 and Cys109 is in dark blue.

**Table 1 brainsci-10-00762-t001:** Metabolic profile of cerebrospinal fluid (CSF) before treatment.

Parameters	CSF Concentrations in Patient	Reference Range
Lactate	0.40	<2.20 mmol/L
Pyruvate	0.05	<0.14 mmol/L
Lactate/pyruvate Ratio	7.9	<20
Acetoacetic acid	0.02	
Beta-hydroxybutyric acid	<0.01	
3-methoxy DOPA	10.5	3–54 nmol//L
3-methoxy-4-hydroxyphenylglycol	19.6	11–46 nmol/L
5-hydroxytryptophane	8.6	3–12 nmol/L
5-hydroxyindolacetic acid (HIAA)	88	63–185 nmol/L
Homovanillic acid	231	156–410 nmol/L
Biopterin	10.8 ↓	14–36 nmol/L
Neopterin	10.2	10–24 nmol/L
**5-methyltetrahydrofolate**	**1** **↓↓↓**	**>44 nmol/L**
Alpha-Interferon	<2	<2 UI/mL

Keys: ↓: decrease; ↓↓↓: strong decrease.

**Table 2 brainsci-10-00762-t002:** List of pathogenicity predictions from different software programs and their relative scores for the pathogenic variant, c.197 G > A, on the *FOLR1* gene.

Software	Version	Score	Prediction
Sift [13]	From dbNSFP3.5	0.02	Deleterious
Polyphen2 hvar [14]	From dbNSFP3.5	0.999	Probably Damaging
CADD phred [15]	1.4	27.5	Correspond to the top 0.0018% of the most pathogenic variants predicted
MutPred2 [16]	2.0	0.952	Damaging
MutationTaster [17]	From dbNSFP3.5	1	Disease causing

**Table 3 brainsci-10-00762-t003:** Prediction of protein instability from different software programs for the pathogenic variant, Cys66Tyr.

Software	Version	Prediction
MUpro [18]	1.1	Decrease stability
CUPSAT [19]	Release 2018.1	Destabilising
AUTO-MUTE [20]	2.0	Decrease stability
Site Directed Mutator (SDM) [21]	Server Access 02/21/2020	Reduced stability
I-Mutant2.0 [22]	2.0	Decrease
mCSM [23]	Server Access 02/21/2020	Destabilising

**Table 4 brainsci-10-00762-t004:** Clinical improvement with folinic acid treatment (based on usual clinical exam).

	Before Treatment	After Treatment (2 Months)
Seizure	**++++**	**+**
Myoclonia	**+++**	▪ less important ▪ less frequent ▪ less invasive ⇨ up to the lack of myoclonia
Tremor	**+++**	**+**
Cognitive functions	Non measurable (due to seizure, myoclonia, fatigue, tremor, language skills, neuromotor skills and limited interaction)	▪ improvement
Language skills	▪ difficulties to speak ▪ no spontaneous question ▪ screaming, laughing and moaning.	▪ speaks more fluently ▪ makes spontaneous questions ▪ makes short sentences.
Motor coordination	▪ motor coordination disorder	▪ better motor coordination: hands reuse (for instance: she holds objects more easily, as a book, turns pages, eats alone, holds a pencil, stacks cubes, switches objects from one hand to another hand and draws on a board)
Neuromotor skills	▪ no autonomous walking ▪ needs a wheelchair ▪ falls down (more than 2 per day)	▪ improvement ▪ walks alone or with partial assistance ▪ stands upright
Interaction	▪ no interaction ▪ no eye contact	▪ interactions around her

**+**: frequency.

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
