# Peer review of "Pharmacoresistant Epilepsy in Childhood: Think of the Cerebral Folate Deficiency, a Treatable Disease"

_brainsci, 2020, doi:10.3390/brainsci10110762_

Round 1

Reviewer 1 Report

The authors report a rare case of cerebral folate deficiency with a FOLR1 homozygous variation. This disease has been described in 2009 by Steinfeld. It is important to report new patients to improve the knowledge of this disease and to allow early diagnosis. Indeed, this disease is treatable and the neurological consequences can be avoided if the treatment starts very early.

Some informations may improve the manuscript. here are some proposals

  • line 59: "The mechanism of these transporters involved in 5-MTHF flux across the blood-brain-barrier is not yet fully understood". Some mechanisms have been proposed for intracerebral transport. Cfr "Choroid plexus transcytosis and exosome shuttling deliver folate into brain parenchyma. Grapp M, Wrede A, Schweizer M, Hüwel S, Galla HJ, Snaidero N, Simons M, Bückers J, Low PS, Urlaub H, Gärtner J, Steinfeld R. Nat Commun. 2013;4:2123. doi: 10.1038/ncomms3123". This information should be changed.
  •  
  • Line 79 : white matter abnormalities have been described. Authors need to improve the description of brain MRI images. The use of the term leukodsytrophy is too broad. Problems of hypomyelination rather than leukodystrophy have been described. In the discussion, a summary of the imaging abnormalities: white matter damage, calcifications, cerebellar damage, ...

Author Response

Reviewer 1

The authors report a rare case of cerebral folate deficiency with a FOLR1 homozygous variation. This disease has been described in 2009 by Steinfeld. It is important to report new patients to improve the knowledge of this disease and to allow early diagnosis. Indeed, this disease is treatable and the neurological consequences can be avoided if the treatment starts very early.

1) line 59: "The mechanism of these transporters involved in 5-MTHF flux across the blood-brain-barrier is not yet fully understood". Some mechanisms have been proposed for intracerebral transport. Cfr "Choroid plexus transcytosis and exosome shuttling deliver folate into brain parenchyma. Grapp M, Wrede A, Schweizer M, Hüwel S, Galla HJ, Snaidero N, Simons M, Bückers J, Low PS, Urlaub H, Gärtner J, Steinfeld R. Nat Commun. 2013;4:2123. doi: 10.1038/ncomms3123". This information should be changed.

Response 1:

Thank you for this very relevant point. We read carefully publications from Grapp et al, Zhao et al, and from Requena Jimenez et al, and we added the following paragraph:

“The mechanism of these transporters involved in 5-MTHF flux across the blood-brain-barrier is not yet fully understood, but some mechanisms have been proposed for intracerebral transport. According to Grapp et al, and to Zhao et al, 5-MTHF binds to FRα at the basolateral side, as it reaches choroid plexus. The 5MTHF-FRα complexes are internalized through receptor-mediated endocytosis. After internalization, the resulting vesicles are translocated into GPI-anchored protein-enriched early endosomal compartments (GEECs). Then, GEECs are transferred to maturated endosomes that are multivesicular complexes (MVC). Intralumenal vesicles (ILV) containing FRα complexes accumulate within MVC and are released as exosomes at the apical side of the epithelium into the CSF, before being endocytosed again by ependyma cells [7,8]. According to Requena Jimenez et al, in this condition, 10-formyl-THF-dehydrogenase (FDH) is crucial, colocalizing with 5-MTHF and FRα within CSF endocytic vesicles. According to in-vitro studies, FDH binds with FRα and is involved in the folate transport regulation by controlling 5-MTHF fluctuations. FDH levels in CSF might also have a key role in the maturation of the leptomeninge arachnoid [9].”

And We added the reference 7,8, and 9 in the “References section”

“7.          Zhao, R.; Aluri, S.; Goldman, I.D. The proton-coupled folate transporter (PCFT-SLC46A1) and the syndrome of systemic and cerebral folate deficiency of infancy: Hereditary folate malabsorption. Mol. Aspects Med. 2017, 53, 57–72.

  1. Grapp, M.; Wrede, A.; Schweizer, M.; Hüwel, S.; Galla, H.J.; Snaidero, N.; Simons, M.; Bückers, J.; Low, P.S.; Urlaub, H.; et al. Choroid plexus transcytosis and exosome shuttling deliver folate into brain parenchyma. Nat. Commun. 2013, 4, 2123, doi:10.1038/ncomms3123.
  2. Jimenez, A.R.; Naz, N.; Miyan, J.A. Altered folate binding protein expression and folate delivery are associated with congenital hydrocephalus in the hydrocephalic Texas rat. J. Cereb. Blood Flow Metab. 2019, 39, 2061–2073, doi:10.1177/0271678X18776226.”

2) Line 79: white matter abnormalities have been described. Authors need to improve the description of brain MRI images. The use of the term leukodsytrophy is too broad. Problems of hypomyelination rather than leukodystrophy have been described.

Response 2:

  • Thank you for your constructive and helpful comments. We added line 79 “A brain MRI was then performed and revealed a severe hypomyelination, according to hypersignals in the subcortical white matter particularly in the cerebellar and subtentorial area predominantly posterior (Figure2).”
  • This is a relevant suggestion. We changed “leukodystrophy” line 79 and 93 by “hypomyelination” and “hypomyelination areas” respectively.

Reviewer 2 Report

This is a fascinating study of a case of cerebral folate deficiency, due to a genetic fault in folate receptor alpha, underlying childhood seizures. This is another example of CFD that responds to high dose folinic acid which bypasses the faulty folate transporter.

There are some errors and omissions in the manuscript that, when corrected, will make the paper acceptable for publication.

  1. Paragraph 2 introduction – the authors refer to food folates being metabolised I the liver to 5 methyl tetrahydrofolate. The metabolic pathway they refer to is for folic acid, an unnatural, synthetic folate which enters the folate cycle via DHFR. Food folate is 5-methyl tetrahydrofolate which is then immediately available for metabolic processes. In its conversion to tetrahydrofolate it donates its methyl group to homocysteine converting it to methionine which then feeds into methylation pathways, while tetrahydrofolate feeds multiple folate metabolic reactions essential for DNA synthesis, neurotransmitter synthesis and others. Thus, both the methylation cycle and synthesis of DNA, key metabolites and neurotransmitters requires folate which then becomes an essential metabolite/vitamin for the brain.

The diagram in figure 1 has an error as a consequence which is that 5 methyl tetrahydrofolate is the main folate transported around the body. The specific pathway for 5mTHF uptake is FR-alpha so that converting folinic acid to 5mTHF would not benefit the system. It is more likely, from other studies, that folinic acid passes across the non-specific pathways of PCFT and RFC directly into CSF where it is immediately available for folate metabolism.

The figure also does not explain the transfer of folate to CSF from the choroid plexus. If as suggested 5mTHF can bypass FRα and diffuse across the membrane into CSF then what is the need for FRα. More likely is that folinic acid transfers across through PCFT and RFC directly into CSF where it is available to cells directly without FRα.

The authors have failed to incorporate the findings of studies demonstrating that FRα transports folate into the CSF in vesicles formed from choroidal membrane. While on the brain side they have not including 10-formyl THF dehydrogenase as a key enzyme acting both as a folate binding protein in CSF and within receiving cells of the brain (see publications of Rameakers and colleagues, Steinfeld and colleagues and Miyan and colleagues)

  1. Introduction paragraph 2 – Folate receptor alpha is not manufactured in the choroid plexus or brain as studies have failed to demonstrate the mRNA for this protein in the CNS or vasculature within the brain. It is thought to be predominantly expressed in the liver and acts as one of the folate binding proteins in the blood and also as a transporter for folate across the choroid plexus into the CSF via vesicular transport.

The case report itself is well presented and the figures are excellent. CSF analysis is particularly enlightening when taken together with the MRI scans.

  1. Discussion: first paragraph need to say which variant does what to FRa functions. The authors say the FRa mutation in this case most likely interrupts folate transport into the brain. Can they suggest how from the structural change?

Para 3: Missing ref for Steinfeld R et al – no date given.

The link to choline metabolism and lack of myelination is well explained.

P6 last para: this reviewer believes that folinic acid is likely to pass across the choroid plexus for direct use by cells and that conversion to 5mTHF prior to transport would not benefit the patient significantly due to the loss of function of FRa.

The point regarding folic acid is excellent and missed by many authors.

P7 para 1 in describing the treatment regime given to the patient the authors do not justify the large dose of folinate given to the patient, particularly the final IV dose of 500mg/week. Were any side effects noted of these doses? Did they try lower doses initially? Is the dose based on other published studies?

Author Response

Reviewer 2

This is a fascinating study of a case of cerebral folate deficiency, due to a genetic fault in folate receptor alpha, underlying childhood seizures. This is another example of CFD that responds to high dose folinic acid which bypasses the faulty folate transporter.

There are some errors and omissions in the manuscript that, when corrected, will make the paper acceptable for publication.

Paragraph 2 introduction – the authors refer to food folates being metabolised I the liver to 5 methyl tetrahydrofolate. The metabolic pathway they refer to is for folic acid, an unnatural, synthetic folate which enters the folate cycle via DHFR. Food folate is 5-methyl tetrahydrofolate which is then immediately available for metabolic processes. In its conversion to tetrahydrofolate it donates its methyl group to homocysteine converting it to methionine which then feeds into methylation pathways, while tetrahydrofolate feeds multiple folate metabolic reactions essential for DNA synthesis, neurotransmitter synthesis and others. Thus, both the methylation cycle and synthesis of DNA, key metabolites and neurotransmitters requires folate which then becomes an essential metabolite/vitamin for the brain.

Response 1:

Thank you for pointing out this mistake. We added to the paragraph 2 in introduction the following paragraph:

“Folate is a family of B vitamin found in human food as folinic acid (5-formyl-THF) converted into 5-MTHF or in synthetic folic acid supplementation. Folic acid is” […]. “Folinic acid can either be immediately available for metabolic processes or be converted into 5,10-methenyl-THF consecutive activities of MTHF synthetase, and MTHF-dehydrogenase [5].”

And We added the reference 5 in the “References section”:

  1. Scaglione, F.; Panzavolta, G. Folate, folic acid and 5-methyltetrahydrofolate are not the same thing. Xenobiotica 2014, 44, 480–488, doi:10.3109/00498254.2013.845705.

The diagram in figure 1 has an error as a consequence which is that 5 methyl tetrahydrofolate is the main folate transported around the body. The specific pathway for 5mTHF uptake is FR-alpha so that converting folinic acid to 5mTHF would not benefit the system. It is more likely, from other studies, that folinic acid passes across the non-specific pathways of PCFT and RFC directly into CSF where it is immediately available for folate metabolism.

The figure also does not explain the transfer of folate to CSF from the choroid plexus. If as suggested 5mTHF can bypass FRα and diffuse across the membrane into CSF then what is the need for FRα. More likely is that folinic acid transfers across through PCFT and RFC directly into CSF where it is available to cells directly without FRα.

Response 2:

Thank you for pointing out this mistake. We changed the figure accordingly.

The authors have failed to incorporate the findings of studies demonstrating that FRα transports folate into the CSF in vesicles formed from choroidal membrane. While on the brain side they have not including 10-formyl THF dehydrogenase as a key enzyme acting both as a folate binding protein in CSF and within receiving cells of the brain (see publications of Rameakers and colleagues, Steinfeld and colleagues and Miyan and colleagues)

Response 3:

Thank you for this very relevant point. We read carefully publications from Grapp et al, Zhao et al, and Requena Jimenez et al, and we added the following paragraph:

“The mechanism of these transporters involved in 5-MTHF flux across the blood-brain-barrier is not yet fully understood, but some mechanisms have been proposed for intracerebral transport. According to Grapp et al, and to Zhao et al, 5-MTHF binds to FRα at the basolateral side, as it reaches choroid plexus. The 5MTHF-FRα complexes are internalized through receptor-mediated endocytosis. After internalization, the resulting vesicles are translocated into GPI-anchored protein-enriched early endosomal compartments (GEECs). Then, GEECs are transferred to maturated endosomes that are multivesicular complexes (MVC). Intralumenal vesicles (ILV) containing FRα complexes accumulate within MVC and are released as exosomes at the apical side of the epithelium into the CSF, before being endocytosed again by ependyma cells [7,8]. According to Requena Jimenez et al, in this condition, 10-formyl-THF-dehydrogenase (FDH) is crucial, colocalizing with 5-MTHF and FRα within CSF endocytic vesicles. According to in-vitro studies, FDH binds with FRα and is involved in the folate transport regulation by controlling 5-MTHF fluctuations. FDH levels in CSF might also have a key role in the maturation of the leptomeninge arachnoid [9].”

And We added the reference 7,8, and 9 in the “References section”

“7.          Zhao, R.; Aluri, S.; Goldman, I.D. The proton-coupled folate transporter (PCFT-SLC46A1) and the syndrome of systemic and cerebral folate deficiency of infancy: Hereditary folate malabsorption. Mol. Aspects Med. 2017, 53, 57–72.

  1. Grapp, M.; Wrede, A.; Schweizer, M.; Hüwel, S.; Galla, H.J.; Snaidero, N.; Simons, M.; Bückers, J.; Low, P.S.; Urlaub, H.; et al. Choroid plexus transcytosis and exosome shuttling deliver folate into brain parenchyma. Nat. Commun. 2013, 4, 2123, doi:10.1038/ncomms3123.
  2. Jimenez, A.R.; Naz, N.; Miyan, J.A. Altered folate binding protein expression and folate delivery are associated with congenital hydrocephalus in the hydrocephalic Texas rat. J. Cereb. Blood Flow Metab. 2019, 39, 2061–2073, doi:10.1177/0271678X18776226.”

Introduction paragraph 2 – Folate receptor alpha is not manufactured in the choroid plexus or brain as studies have failed to demonstrate the mRNA for this protein in the CNS or vasculature within the brain. It is thought to be predominantly expressed in the liver and acts as one of the folate binding proteins in the blood and also as a transporter for folate across the choroid plexus into the CSF via vesicular transport.

Response 4:

We carefully checked this point and corrected it by “In the central nervous system, different folate transport systems are found, including the reduced folate carrier (RFC), the proton coupled folate transporter (PCFT) and the receptor of folate alpha (FRα). FRα represents the higher affinity folate transport system compared to RFC and PCFT”.

The case report itself is well presented and the figures are excellent. CSF analysis is particularly enlightening when taken together with the MRI scans.

Response 5:

Many thanks for this comment.

Discussion: first paragraph need to say which variant does what to FRa functions. The authors say the FRa mutation in this case most likely interrupts folate transport into the brain. Can they suggest how from the structural change?

Response 6:

Thank you for this very relevant point. The variant observed in this study is NM_016724.2. We added “(NM_016724.2)” in the manuscript.

We thank the reviewer for his relevant question. We acknowledge that the mutation of Cys 66 to Tyr leads to the absence of disulfide bond which, in turn, significantly modify the conformation of FRα. This may lead to an increase of two domains distance (Cys66 is located on the second β-strand (pos 63-67) and Cys109 is located on the fourth α-helix (pos 94-109) (Figure 4)). However, the impact to FRα function is not obvious since Cys66-Cys109 disulfide bond is located far away from the functional site. We hypothesize that the Cys66Tyr mutation mostly affects protein dynamics which in turn may modulate the function of FRα. But, this hypothesis needs to be validated by different structural and dynamical techniques (such as computational chemistry/molecular dynamics simulation) and in vitro experiments.

We added the following paragraph:

“Grapp et al., [1], have suggested that missense mutations might lead to an instable conformation of FRα explained by a loss of a structurally important disulfide bond or disturbed post-translational processing. Splice and stop mutations might result respectively in a truncated FRα and in a premature termination codon and subsequent complete lack of FRα expression. Duplications are likely to cause major misfolding leading to premature degradation of FRα [13]. […]

In our case the mutation of Cys 66 to Tyr lead to the absence of disulfide bond which, in turn, is likely to significantly modify the conformation of FRα. This may lead to an increase of the two domains distance (Cys66 is located on the second β-strand (pos 63-67) and Cys109 is located on the fourth α-helix (pos 94-109) (Figure 4)). However, the impact to FRα function is not obvious since Cys66-Cys109 disulfide bond is located far away from the functional site. Thus, we hypothesize that the Cys66Tyr mutation mostly affects protein dynamics which in turn may modulate the function of FRα.”, and we removed “Our variation induces a lack disulfide-bond between the Cys66 and Cys109 residues”.

Para 3: Missing ref for Steinfeld R et al – no date given.

Response 7:

Thank you for pointing out this mistake. We added the reference.

The link to choline metabolism and lack of myelination is well explained.

Response 8:

Many thanks for this comment

P6 last para: this reviewer believes that folinic acid is likely to pass across the choroid plexus for direct use by cells and that conversion to 5mTHF prior to transport would not benefit the patient significantly due to the loss of function of FRa.

Response 9: According to the first point (Response 1), we modify the sentence :” Supplementation with folinic acid have been reported to be efficient with a rapid reverse of neurological symptoms and an improvement of brain functions [10,13,17,18] [10,16,17]. This molecule is rapidly converted after several enzymatic steps to a biological active 5-MTHF in the body” by: “Supplementation with folinic acid have been reported to be efficient with a rapid reverse of neurological symptoms and an improvement of brain functions. This molecule is known to pass across the non-specific pathways of PCFT and RFC directly into CSF where it is immediately available for folate metabolism [10,13,17,18].

The point regarding folic acid is excellent and missed by many authors.

Response 10:

Many thanks for this comment

P7 para 1 in describing the treatment regime given to the patient the authors do not justify the large dose of folinate given to the patient, particularly the final IV dose of 500mg/week. Were any side effects noted of these doses? Did they try lower doses initially? Is the dose based on other published studies?

Response 11:

Thank you for this very relevant point. There is no side effect observed currently. We added “with no side effect observed” in the discussion. We did not try lower dose and arbitrary choose this efficient dose.

We wish to thank the reviewers for their constructive and helpful comments. Nevertheless, do not hesitate to contact us for further explanations/modifications if necessary.

I am looking forward to hearing from you.

Sincerely,

Dr. Pierre Antoine FAYE,

PharmaD, PhD

This manuscript is a resubmission of an earlier submission. The following is a list of the peer review reports and author responses from that submission.